# Syringe Access, Syringe Sharing, and Perceptions of HCV: A Qualitative Study Exploring the HCV Risk Environment in Rural Northern New England, United States

**DOI:** 10.3390/v16091364

**Published:** 2024-08-27

**Authors:** Eric Romo, Elyse Bianchet, Patrick Dowd, Kathleen M. Mazor, Thomas J. Stopka, Peter D. Friedmann

**Affiliations:** 1Department of Population and Quantitative Health Sciences, University of Massachusetts Chan Medical School, Worcester, MA 01655, USA; 2Office of Research, University of Massachusetts Chan Medical School—Baystate, Springfield, MA 01199, USA; 3Department of Public Health and Community Medicine, Tufts University School of Medicine, Boston, MA 02111, USA

**Keywords:** rural, hepatitis C virus, risk environment, injection drug use, harm reduction, syringe sharing

## Abstract

The ongoing hepatitis C virus (HCV) epidemic in the United States disproportionately affects rural people who inject drugs (PWID). This study explores the HCV risk environment in rural northern New England by examining PWID experiences and perceptions of HCV and injection equipment-sharing practices. We performed a thematic analysis on semi-structured interviews conducted with 21 adults with a history of injection drug use from rural New Hampshire, Vermont, and Massachusetts between April 2018 and August 2019. Salient themes included: (1) limited and varied access to sterile syringe sources; (2) syringe scarcity contributing to the use of informal syringe sources (e.g., secondary syringe exchange or syringe sellers who purchased syringes from out-of-state pharmacies); (3) syringe scarcity contributing to syringe sharing; (4) linkages among decisions about syringe sharing and perceptions of HCV risk, HCV status, and interpersonal trust; and (5) confusion and misconceptions about HCV, including difficulty learning one’s HCV status, inadequate HCV education, and misconceptions regarding HCV transmission and treatment. Efforts to prevent and eliminate HCV among rural PWID should expand syringe access, increase awareness of HCV as a serious but preventable risk, and acknowledge social connections as potential influences on syringe access and syringe-sharing decisions.

## 1. Introduction

The intersecting epidemics of opioid use disorder, injection drug use, and hepatitis C virus (HCV) infection have disproportionately affected rural communities in the United States. Between 2010 and 2019, the number of new HCV infections in the U.S. increased nearly four-fold [1], with new infections occurring predominantly among young people who inject drugs (PWID) living in rural areas [2,3]. Most of these new HCV infections are likely attributable to injection-sharing practices, including sharing syringes and other drug paraphernalia [3,4,5].

The risk environment framework posits that at least four types of environmental factors—physical, social, economic, and policy—work together to influence the risk of drug-related harms [6,7,8]. Therefore, any effort to prevent and eliminate HCV infection among a population of rural PWID requires an understanding of the structural and environmental factors that affect HCV-associated behaviors in this setting. Since syringe sharing is the most important risk factor associated with HCV transmission among PWID and one of the most important risk behaviors associated with HIV transmission in this population [9], it is important to understand the factors that impact access to sterile syringe sources, such as syringe services programs (SSPs) and pharmacies that offer nonprescription syringe sales. The 2015 HIV outbreak among PWID in Scott County, Indiana, USA, illustrated that high rates of HCV can be a reflection of extensive transmission networks and a potential harbinger of future HIV outbreaks [10,11].

Given that factors contributing to the HCV risk environment can be complex and context-specific, this environment is often best elucidated using qualitative methods. Recent qualitative studies have begun to characterize the HCV risk environment among rural PWID. In a study from rural Illinois [12], PWID and key informants described an environment characterized by (1) economic instability that increased psychosocial distress and worsened drug use; (2) poor physical accessibility to specialty providers for substance use disorder (SUD) and HCV treatment; (3) policies rooted in stigma that made it challenging for PWID to purchase sterile syringes at pharmacies; and (4) a social environment rife with stigma from healthcare and SUD providers, which contributed to misconceptions about HCV transmission and treatment. In a study from rural Appalachian Kentucky [13], young adult PWID described similar features of the HCV risk environments, adding that fear of police exposure drove local PWID to inject in “trap houses”, where injection equipment-sharing practices were common, and describing a social environment where HCV was perceived as ubiquitous. This perception of HCV as ubiquitous or “inevitable” has also been observed among PWID from rural West Virginia, Kentucky, and North Carolina, the majority of whom were not concerned about acquiring or transmitting HCV [14]. Policy barriers to purchasing syringes at pharmacies have been particularly well documented among rural PWID. PWID in Kentucky and New Hampshire have reported difficulty finding pharmacies willing to sell syringes, as well as restrictive pharmacy policies, including the requirement to show identification or to have a prescription or proof of medical need for a syringe [15,16].

Few qualitative studies have explored the HCV risk environment among PWID in rural New England. In recent years, the New England region has experienced a surge in hospitalizations for injection drug use-associated infective endocarditis [17] and clusters of HIV infections among PWID [18] and contains several counties that the Centers for Disease Control and Prevention (CDC) has identified as vulnerable to the rapid spread of HIV and HCV infections among PWID [11]. An understanding of the HCV risk environment in rural New England is needed to help inform local harm reduction interventions. Drawing on in-depth qualitative interviews with rural adults with a history of injecting drugs living in northern New England, we explored rural PWIDs’ experiences with and perceptions of acquiring injection supplies, injection equipment sharing practices, and HCV.

## 2. Methods

### 2.1. Data Source

The Drug Injection Surveillance and Care Enhancement for Rural Northern New England (DISCERNNE) study was a mixed-methods, cross-sectional study among people who use drugs (PWUD) conducted in 11 rural counties along the Connecticut River Valley in New Hampshire (NH), Vermont (VT), and Massachusetts (MA). Using surveys, qualitative interviews, and infectious disease testing, the overall aim of DISCERNNE was to characterize the risk environment and epidemiology of overdose and injection-mediated infectious diseases among rural PWID. DISCERNNE is part of the Rural Opioid Initiative (ROI), a consortium of eight studies across 10 states and 65 U.S. counties developed to characterize and better inform interventions for the opioid crisis in geographically diverse rural regions of the United States [19].

### 2.2. Participant Recruitment

Eligibility criteria were the following: (1) age 18 years or older, (2) spent most of the last 30 days living in the study area, (3) used opioids “to get high” or injected any drug in the last 30 days, and (4) were able to provide written informed consent. Respondent-driven sampling was used to recruit participants across 11 study sites that were selected after consulting local public health officials and service providers. DISCERNNE enrolled 589 participants, and 22 completed in-depth, semi-structured interviews between April 2018 and August 2019. Most interviewees (n = 15) were recruited from among those who had participated in the quantitative survey component of DISCERNNE, while the remainder were recruited through street outreach and participant referral. For qualitative interviews, purposive sampling enrolled a sample reflective of the local drug-using community by sex, age, and opioid use patterns. The present analysis excluded one participant who reported never having injected drugs. The final analytic sample of 21 participants includes those who reported currently injecting drugs (past 30 days, n = 17) and those who reported previously injecting drugs during the past year (n = 4). The Baystate Health Institutional Review Board approved the study protocol (IRB# 1094092).

### 2.3. Data Collection

The Qualitative Working Group of the ROI consortium, with representatives from DISCERNNE and the other ROI studies, developed a core interview guide for semi-structured interviews. These guides were then further adapted to individual study sites. The risk environment framework informed the development of the DISCERNNE interview guide. Domains assessed included a personal narrative of substance use; experiences and beliefs about injection and sexual behaviors; interactions with law enforcement; perspectives on local laws and policies; and experiences and perspectives on local health care, addiction treatment, and syringe exchange services. The interview guide was pre-tested by senior members of the research team. The final interview guide consisted of 35 questions, though not all were asked at every interview.

Participants provided written informed consent and then engaged in 45–90 min, in-depth, semi-structured interviews. Interviews were conducted in person at DISCERNNE study sites (co-located with or in proximity to local harm reduction agencies) between April 2018 and August 2019. Interviewers (TS, ER, PF, and others who were not co-authors) included male and female epidemiologists, physicians, doctoral-level graduate students, and public health specialists with many years of substance-use-focused research experience and extensive prior training and experience with qualitative interviewing. All interviews were audio recorded and transcribed verbatim. Participants received USD 25 for their time. Of the 21 interview participants, 15 also participated in the survey component of DISCERNNE. This subset of participants provided more detailed self-reported demographic information. This study adhered to the COnsolidated criteria for REporting Qualitative research (COREQ). A completed COREQ checklist is included in the Appendix A.

### 2.4. Data Analysis

This qualitative analysis builds on a preliminary analysis of the DISCERNNE in-depth interviews. A five-member coding team previously coded the interview transcripts for a range of predetermined and emerging salient topics. This first stage of coding entailed a deductive approach, applying codes based on qualitative interview domains and items focused on the acquisition of injection supplies, injection sharing practices, and HCV—the focus for our analysis.

Using a thematic analysis approach [20], a three-member research team conducted an in-depth review and analysis of the interview transcripts with respect to these topics. The research team first developed a preliminary coding scheme based on the primary areas of interest and emergent topics that arose from a close reading of the relevant coding reports from the original analysis. Each member of the coding team coded each transcript independently. Transcripts were coded in their entirety, with special attention paid to portions of the text that were assigned codes relevant to our topics of interest during the original analysis. The research team met weekly to discuss and resolve any discrepancies in coding, refine existing codes, and make any necessary revisions to the coding scheme. The final codebook contained 13 parent codes and 25 child codes. The research team reviewed the resulting coding reports and memos, organized the data into preliminary themes, and held regular discussions until they reached a consensus on the final themes. Qualitative software (Dedoose 8.2, Los Angeles, CA, USA) was used to facilitate the management and coding of interview transcripts. To protect confidentiality, participants’ names were replaced with pseudonyms.

## 3. Results

The sample included 11 females (52%) and 10 males (48%), with ages ranging from 23 to 55 years (median age: 29.5 years) (Table 1). As already noted, 17 participants (81%) reported currently injecting drugs (past 30 days), and the remaining 4 (19%) reported previously injecting drugs during the past year. Eleven participants (52%) lived in VT, six (29%) lived in NH, and the remaining four (19%) lived in MA. Among the 15 participants who provided more detailed demographic information on the survey component of DISCERNNE, 14 (93%) identified as non-Hispanic White, 12 (80%) had at least a high school education, 8 (53%) experienced homelessness in the previous 6 months, and 6 (40%) were incarcerated in the previous 6 months. Of this subset of 15 participants, 12 (80%) reported heroin as their drug of choice, and 3 (20%) reported fentanyl/carfentanil as their drug of choice. Twelve participants (57%) were HCV seropositive. For most participants (n = 15), HCV serostatus was determined using a rapid antibody test administered as part of the larger DISCERNNE study. The remaining participants self-reported their HCV status.

Five themes were generated from the data: limited and varied access to sterile syringe sources; syringe scarcity contributing to the use of informal syringe sources; syringe scarcity contributing to syringe sharing; linkages among decisions about syringe sharing, and perceptions of HCV risk, HCV status, and interpersonal trust; and confusion and misconceptions about HCV. Figure 1 provides a visual summary of our findings.

### 3.1. Limited and Varied Access to Sterile Syringe Sources

Participants’ descriptions of the physical and policy environment with respect to SSP access were consistent with the geographic differences in SSP capacity in our study area. At the time of our study, only five fixed-site SSPs were operating in our 11-county study area: four across six VT counties, one in the sole MA county, and none across four NH counties (where SSPs were only legalized in 2017). Accordingly, several participants from MA and VT reported using a nearby SSP regularly, and participants living in NH knew of no nearby SSPs. One participant, a 26-year-old female (26F, NH, HCV+), complained about the lack of SSPs after returning to NH a year before:

I was hoping a little progression had been made … like there were no needle exchanges before. There’s still none now. (Mary, 26F, NH, HCV+)

Although several participants from MA and VT reported using SSPs, many from VT described a policy environment that created barriers to accessing existing SSPs. The most prominent policy barrier was limited hours of operation. The four SSPs in our VT study counties were only open once or twice per week for a total of three to eight hours. Compared to the SSP in MA county, which was open five days per week for a total of 40 h. Participants had only a brief, often inconvenient, window of time to exchange syringes:

They [are] open only twice a week. And sometimes you can’t get here. You know, I’m down here every day, but still the time which is short, it’s only like an hour and half. (Susan, 55F, VT, HCV+)

Two participants from different VT towns explained that they and others were uncomfortable attending their closest respective SSP because of its location near a health clinic.

Two of my smaller children, my younger children[’s] doctor’s office is upstairs, so I wasn’t sure of coming to the exchange. (Lisa, 28F, VT, HCV+)

I know that this specific location makes it kind of difficult just because [buprenorphine provider’s clinic] is right there … I know lots of people that would like to come here but they’re too nervous. (Karen, 33F, VT, HCV−)

Participants also reported variable access to pharmacies offering nonprescription syringe sales. Although participants from all three states reported purchasing syringes from a local pharmacy, several from across the study area described living a long distance from the nearest pharmacy willing to sell syringes. In some cases, this was simply due to living in a very rural town without any pharmacy at all, as John describes: “There’s no gas station. There’s no nothing …. If you live in [Town], you travel.” (29M, MA, HCV+). In most cases, however, this was because nearby pharmacies were unwilling to offer nonprescription syringe sales.

In Connecticut you can buy them in bulk because if you can bat your eyes enough, you can get them from a pharmacy … so it basically depends on where you are. Up here I’m struggling to find them … I’ve got family in Connecticut so … when I took trips I’d stock up. (Mary, 26F, NH, HCV+)

### 3.2. Local Syringe Scarcity Contributes to the Use of Informal Syringe Sources

Participants living in risk environments marked by limited local access to SSPs and/or pharmacies were left to obtain syringes from other sources. In VT, several participants described obtaining syringes indirectly from an SSP via someone in their social network (i.e., secondary syringe exchange). Jennifer (38F, VT, HCV−) had a sister who was willing to travel to an SSP in a neighboring town to obtain clean syringes and other injection supplies on her behalf.

And then my sister does the needle exchange in [Town], and she gets [Narcan] somewhere over there. And she stocks up for the both of us. So once a week she comes from [Town], or [Town], and brings me all the stuff, I guess to try to keep me safe. (Jennifer, 38F, VT, HCV−)

As noted in a previous quote, Lisa (28F, VT, HCV+) was uncomfortable attending the nearby SSP because of its proximity to her children’s doctor’s office. She relied on her fiancé to obtain supplies from the SSP on her behalf: “[M]y ex-fiancée used to come…and he would get, you know, we would have a fresh like, you know, new needles and stuff”.







Participants in our NH study counties, who had no local SSP access and limited in-state pharmacy access, explained that PWID in NH relied heavily on local people who sold drugs/syringes who purchased syringes from distant or out-of-state pharmacies and sold them locally for a profit (USD 3 to 5 per syringe).

The dealers here that were going to [town in MA]. It would be one of their stops on the way back or on the way there, they’d grab the needles and so then when they got back, they could sell their drugs and their needles, and, you know, times their profit by five on the needles. (Ken, 24M, NH, HCV−)

We have to go to Massachusetts or Vermont or somewhere to get the needles. And then we’ll bring them back, and either people will sell them for $5 a pop or something. They can make 50 bucks on a bag of 10 needles. You know what I mean? (Julia, 23F, NH, HCV−)

However, one participant reported that many in NH “don’t want to sell [syringes] because they don’t know … the next time they can get a bag of them”, suggesting that local syringe sellers who rely on out-of-state pharmacies may not be reliable syringe sources for local PWID.

### 3.3. Syringe Scarcity Contributes to Syringe Sharing

Due to the high level of syringe scarcity in our rural northern New England study area, most participants reported sharing a syringe with another PWID at some point in time. Several described sharing during times of acute syringe scarcity. Participants were willing to share a syringe with someone else if it meant avoiding or alleviating withdrawal symptoms: “I did use a used one…’cause I just wanted to get right, you know?” (Ken, 24M, NH, HCV−).

One participant, Ben (32M, NH, HCV+), drew a direct line between syringe sharing and the chronic syringe scarcity caused by a lack of access to nonprescription syringe sales. He blamed efforts to limit syringe access for increasing syringe sharing and the spread of infectious diseases:

And they think that because if they let the pharmacies sell needles that it’s just going to get worse and worse and worse, but it’s not stopping us from using. It’s not. I mean in all reality it’s,—we’re just spreading disease. Sharing dirty needles. Because we’re going to use no matter what. Just because CVS doesn’t sell needles doesn’t mean we’re not going to use a dirty one. So, I think that’s important for people to know. They’re not stopping us from using because they don’t sell needles. (Ben, 32M, NH, HCV+)

Ben and one other participant described PWID becoming so desperate to relieve withdrawal symptoms (i.e., feeling “dope sick”) that they were willing to use discarded syringes:

Ben:[I]f somebody wants to get high and they don’t have a needle to use, they’ll pick one up off the ground and use it. That’s how desperate they are.

Interviewer:So there are people in that situation where they don’t know who used it before them?

Ben:It could’ve been 500 people and they’ll still pick it up and use it…They’ll hunt it down until they find it. And it doesn’t matter where they find it. Cause I’ve, I’ve done it before. (Ben, 32M, NH, HCV+)

When you’re an addict … and you’re high or sick and you ain’t got one, you’re gonna use it. You’re gonna find one. You know, I’ve picked them up off the street and gone underneath the railroad bed … and found one, [brought] it back to the house. I didn’t clean it, I just used it…I was too sick…I didn’t give a hoot. (Larry, 47M, NH, HCV+)

Ultimately, participants from across the study area described social environments where syringe sharing was common, and PWID perceived syringe-sharing behavior and related infectious diseases as normative, if not ubiquitous.

And I think [SSP] is probably saving a lot of people from spreading stuff the way people share needles around here. (Nancy, 29F, VT, HCV+)

And it is crazy out here the amount of people, if you had a line of all the needle junkies around here, 98% of them have Hep C, at least. They don’t care about sharing needles around here. It’s like nobody around here is scared of what can happen until it’s happening… Cause everybody I ask or everybody I’ve ever shot up with, I ask them if they have hep. “Yep, I have hep. I got three strands of it. Oh, I got two strands of it”. (Jason, 28M, VT, HCV+)

### 3.4. Linkages between Decisions about Syringe Sharing and Perceptions of HCV Risk, HCV Status, and Interpersonal Trust

For interview participants, syringe sharing and HCV were connected and frequently mentioned together, as illustrated by the previous quote from Jason. Participants’ decisions about whether, with whom, and under what circumstances to share syringes were strongly influenced by features of their social environment, namely perceptions of HCV risk, their own HCV status, and interpersonal trust. Participants described a range of syringe-sharing behaviors among PWID: those who did not share syringes, those who shared syringes regularly, and those who only shared under certain conditions.

#### 3.4.1. Did Not Share Syringes

At one extreme were participants who did not share syringes at all. Most cited fear of acquiring an infectious disease, especially HCV and HIV, as their reason for not sharing. Others said they stopped sharing injection equipment after learning they were HCV positive out of concern for spreading HCV to others. Notably, all participants who reported they did not share syringes also reported being able to obtain clean syringes from either an SSP or a nearby pharmacy.

I use my own [needles]. I have been since I learned that I have contracted Hepatitis C. I use my needles, my needles only, clean ones come, I see come out of a package. I buy them myself. (Mark, 24M, NH, HCV+)

I do not share. I don’t share needles, I don’t share cottons, I do not share waters or what I mix my stuff in because I know that I’m Hep C positive. (Lisa, 28F, VT, HCV+)

One of these participants, Mark, was so concerned about spreading HCV to others, he went so far as to burn his used syringes to ensure that no one else would use them.

Interviewer:Where do you dispose of the syringes?

Mark:[M]ost of the time I burn them…to make sure that nobody touches it especially knowing I have Hepatitis C. I burn them now.

Interviewer:Where do you burn them?

Mark:[I]n my fire pit in my house…throw the spoons in it because I believe that if I throw the spoon in there if there was anything on that spoon at, you know…so many hundred degrees I believe that it would be fine. (Mark, 24M, NH, HCV+)

#### 3.4.2. Shared Syringes Regularly

At the other extreme was PWID, who shared syringes frequently. Several participants described other PWID, often in negative terms, who were indifferent about the risk of acquiring HCV from syringe sharing.

[W]hen I was living in [City in MA], sometimes people would knock on the door at night and ask me for a needle. And I’d say hey, I used it. I have Hep C. They’d say I don’t care. They’re puking up off the side of the railing. They don’t care about Hep C. You know, they want that fix…You know, people just don’t care about getting Hep C. If they’re sick, they just want to use the needle and get better. (John, 29M, MA, HCV+)

[B]efore I found out that my ex-fiancé had it, I had even encouraged him, like before I had started using, you know, don’t share anything. Like it’s very important not to. But he’s like “I don’t know what the big deal is”, and he ended up with two strains of Hep C. (Lisa, 28F, VT, HCV+)

One participant strongly believed this indifferent attitude toward HCV arose from PWID perceiving HCV as curable:

[L]ike so people now think like, these kids think that aw, screw it. If I get it, I can just take the medicine and it will go away. … That’s why I’m seeing, I see like people just sharing all the time. It’s mind boggling. And they just don’t even give a shit. Like I hadn’t heard too much about HIV around all that much for some reason but Hep C like everybody pretty much has it. It’s because they know they can get rid of it for some reason. Like ever since that drug came out or that medicine, oh my god people just started … they don’t care at all. (Brian, 39M, MA, HCV−)

Although acquiring HCV inspired some participants to stop sharing syringes, for others, an HCV diagnosis prompted a fatalistic attitude toward syringe sharing:

And then, after [acquiring HCV] I was like well, I already got it. I didn’t know there was that many strains at first and then this girl says oh yeah, there’s friggin seven strains or something like, or six, or five, or something like that … So, I probably got all kinds of it now. But yeah … I shared with my girlfriend. I probably shared with a dozen different people. And that’s a low number compared to some people around here. (Jason, 28M, VT, HCV+)

Well, I avoid sharing because I have Hep C. Some people say “Oh, well I already got it”, so it doesn’t matter to them. (John, 29M, MA, HCV+)

#### 3.4.3. Shared Syringes under Certain Risk Conditions

Between those two extremes were PWID who were willing to share syringes, but only under certain risk conditions. For example, two participants said they would only inject with a used syringe after cleaning it with bleach, something one of them suggested was an “old school” harm reduction method not practiced by less experienced PWID.

More commonly, participants tried to mitigate the risk of syringe sharing by only sharing with certain people whose status they believed they knew.

I’d have to know them for a while. I’d have to know for a fact they don’t have HIV because I am not, you know … As long as I know that they, you know, I know them, you know, very well and they’re not considered what I would consider a dirty person. I mean like not ever showering, not using deodorant, and not nasty crap. Um, then I’d consider it, but there’s not a lot of people like that around here. (Jason, 28M, VT, HCV+)

We do use our needles sometimes together. But we won’t use it with anybody else. Just the family…just my kids. I won’t use anybody else’s or nothing. Nope. I don’t know if they have AIDS. I don’t know anything. I know what my kids have…And I know they don’t have AIDS. They have hepatitis, just like me, so I don’t really matter on that. (Susan, 55F, VT, HCV+)

I used a used needle, from a friend, from a friend of mine, from a friend of ours, me and my girlfriend, who I trusted. And she, you know, she claimed and promised that she was the only one that used it and that she was clean. And I wanted to get right. So I told myself that I could believe her, and, uh, to this day I’m still test negative for any diseases. But I did use a used one, and I took her word for it cause I just wanted to get right, you know?’ (Ken, 24M, NH, HCV−)

These quotes illustrate that a participant’s trust in someone else’s infectious disease status, and thus their willingness to share syringes with them, was informed by the intimacy of their relationship. Several participants described only sharing syringes with family members or romantic partners. One participant only shared syringes with his girlfriend and considered the idea of sharing syringes with his friends to be “disgusting”. As the above quote from Jason shows, some participants also associated infectious disease risk with personal hygiene and cleanliness.

One participant explained that the PWID in his small town all knew each other and, therefore, were more likely to trust one another’s infectious disease backgrounds and share syringes with one another:

See the other thing is that being in this area here and it being, being so small of a place, that you basically know the person that you’re using with…That person that we know them, we know their background, we know their history. … And basically, what people say to each other is “What do you got? You got Hep C?” And I’ll say “Yeah”, she goes “Well me too”. “That’s it? That’s all you got is Hep C?” “Yeah, that’s all I”, “Oh I got Hep C too. All right”. So, we’ll use, and we’ll share the same syringe. (Larry, 47M, NH, HCV+)

This quote also depicts how HCV status can influence with whom someone is willing to share syringes. Some participants with HCV, like Larry, were willing to share syringes with others whom they trusted “only” had HCV.

Notably, some participants recognized that the strategy of only sharing syringes with people who meet certain criteria has its shortcomings. A couple of participants realized that the infectious disease status of their sharing partners could unknowingly change over time, or sharing partners could lie about their disease status.

It’s funny now that you say it, now that you think about it. How do you pick who you choose to share with? That is kind of fucked up because you don’t know what’s in their body. They could be lying to you…that’s scary to think about. They could be lying to you because I’ve lied to people before to get free shit. (Jason, 28M, VT, HCV+)

### 3.5. Confusion and Misconceptions about HCV

Participants described a social environment marked by confusion and misconceptions about HCV. Although perceived HCV risk and HCV status played an important role in syringe-sharing decisions among participants, several reported having difficulty learning their HCV status. There were participants who reported previously being told they were positive for HCV antibodies who remained unsure what their status was. These participants remained confused despite multiple reported contacts with health care clinicians.

Well, I think the first time I found out was, I was in [rehab] over in [Town]…Six years ago. So, right there and then I get tested when I got out at my regular physician ’cause I told them, and they test me and said I don’t have it. So, I’m really confused if I do, I don’t. One said I do. But some said that you can, that the antibodies can clear up. So, I don’t know. I really don’t know. I just say I do. Because all my liver tests come out high every time. (Susan, 55F, VT, HCV+)

Nancy:I have Hep C. I, well I have the antibodies for it. I don’t know. I got tested after when I was pregnant, and they actually didn’t tell me. It was in the NICU’s nurse’s notes. Mother Hep C positive. And I was reading it one night, and we saw it. And then she looked in her computer and showed me, but I haven’t, I think the prison checked my levels once but they never told me what they were.

Interviewer:So, you don’t know exactly what your Hep C status is?

Nancy:No. No. (Nancy, 29F, VT, HCV+)

Some participants recounted that when they were told their HCV test results, they did not receive adequate education about HCV. Other participants reported receiving little, if any, education about HCV treatment options during healthcare encounters. Participants’ narratives suggested health care providers often did not give the impression that an HCV diagnosis was serious or urgent or were uncomfortable discussing the findings and their implications with patients.

Interviewer:[H]as anybody in this process of testing, saying yes you do, no you don’t, has anybody kind of sat down with you, talked about what it is exactly and treatment?

Susan:No…

Interviewer:You haven’t talked to a provider about any kind of treatment and…

Susan:No…They sent me a letter saying that I should cause my liver function’s high. Um, my regular physician said I should go get [tested]…but I had already done it…That’s all my provider said. She hasn’t sat down and talked to me about any of it. Nobody has. (Susan, 55F, VT, HCV+)

Receiving inadequate HCV education, participants often had misconceptions regarding HCV transmission and treatment. For instance, many PWID were unaware that HCV could be transmitted by sharing injection equipment other than syringes:

Interviewer:So, have you had any experiences of sharing paraphernalia or needles with other people?

Mark:Um, not needles but the same spoon. And that’s where I believe I contracted Hepatitis…So I wasn’t aware of the fact that that could be transmitted like that…Like that needle had been in his arm you know, thinking about that, the needle had been in his arm. I mean he had Hep C and then he put it in the thing, and I put mine in there and you don’t think about that. Ever. I don’t know why. I, it just dawned on me when he told me about it. (Mark, 24M, NH, HCV+)

Although MA, VT, and NH removed sobriety requirements for HCV treatment in their Medicaid programs in the few years prior to the study period, there were participants who still believed they had to maintain abstinence from drug use before they could receive HCV treatment:

I have Hepatitis C. I’ve had it, God I can’t even remember, and I haven’t been clean long enough to go through the treatment to get rid of it. You need to have like six months clean, and, um, if you get it back again, they don’t want to give you the treatment. (Olivia, 32F, NH, HCV+)

## 4. Discussion

This study explored experiences with and perceptions of acquiring injection supplies, injection equipment sharing practices, and HCV among 21 PWID to examine aspects of the HCV risk environment and their impact on syringe sharing in rural northern New England. Overall, participants described a physical and policy environment characterized by limited access to syringe sources in much of rural northern New England, especially in NH. This risk environment marked by local syringe scarcity drove PWID to seek syringes from informal sources and engage in syringe sharing. However, syringe sharing was more than a product of the availability of syringe sources; features of PWID social environments—including perceptions of HCV risk, their own HCV status, and interpersonal trust—also influenced decisions about whether and with whom to share injection paraphernalia. Despite HCV playing an important role in syringe-sharing decisions, participants often described a social environment characterized by confusion and misconceptions regarding the transmission, diagnosis, and treatment of HCV.

Our results illustrate the importance of understanding the unique context of rural PWID, even across demographically and geographically similar populations, when considering their needs and designing harm reduction interventions. Our sample of rural PWID was not monolithic. Participants described three different risk environments with respect to SSP access: rural NH, where SSPs were unavailable; rural VT, where SSPs existed but could be difficult to access; and one rural MA county, where an SSP operated nearly 40 h per week. The difference in SSP access among rural MA, NH, and VT could be the consequence of different state policies and physical environments. During the study period, VT provided less robust state funding for SSPs than MA, and NH provided no state funding for SSPs [21]. There was a similar variation in access to nonprescription syringe sales. At the time, MA, NH, and VT all permitted the nonprescription sale of syringes at pharmacies, but pharmacy participation was, and remains, optional. Pharmacists were often unwilling to sell syringes to PWID because of stigma toward PWID and a belief that syringe sales were more harmful than beneficial [15,22]. While nearly all pharmacies in MA offered nonprescription syringe purchases [23], recent qualitative research suggested that PWID in rural NH had very limited pharmacy access to syringes [16]. Consistent with this research, only participants from NH and VT described living near a pharmacy that was not willing to sell syringes. PWID in rural Illinois and Kentucky have reported similar barriers to purchasing syringes at pharmacies [12,13]. However, participants in our study also shed light on a barrier unique to rural PWID: a lack of access to nonprescription syringe sales simply because one lives in a small town without a pharmacy at all.

As observed in other rural studies [12,13], we found living in a risk environment characterized by local syringe scarcity drove many PWID to obtain syringes from informal sources. Most notably, participants in NH relied heavily on local people who sold drugs and bought syringes in bulk at out-of-state pharmacies. One other qualitative study among PWID observed the same phenomenon in a different rural region of NH [16], and a prior urban northeastern study reported on-street syringe sellers of various backgrounds [24]. The participants in the NH study added that interstate travel carried significant risk from law enforcement, who frequently pulled over PWID to search for drugs or drug paraphernalia. The fact that people who sold drugs in our study area were willing to take on additional risk to procure syringes at faraway pharmacies, and local PWID were willing to pay significantly marked-up prices for those syringes, reflects the level of unmet need for sterile syringes in several rural NH communities. Our results reiterate that rural PWID are willing to confront the significant distance and transportation barriers to sterile syringes, but doing so incurs costs (e.g., time and/or money) and may expose PWID to other unanticipated risks. Any harm reduction interventions aimed at rural PWID must consider such nuances of the local risk environment.

Our results illustrate that syringe access is situated at the intersection of the physical, policy, and social environments in rural communities. We found that many PWID relied regularly on their social network of friends and family for sterile syringes. In particular, secondary syringe exchange enabled PWID to overcome barriers to SSPs, including distance and fear of public exposure. Similar findings have been observed among urban PWID [25,26,27,28]. Although urban PWID generally have greater spatial access to SSPs than rural PWID, secondary syringe exchange is a common practice in these settings [24,29]. Like those in our study, urban PWID have cited fear of exposure, fear of police harassment, outstanding warrants, poor health, and inconvenient and inadequate SSP hours as barriers to directly attending SSPs [25,26,27]. Future interventions could consider training peers or engaging existing secondary exchange networks to deliver harm reduction materials and education.

Our study provides several important implications for HCV prevention. Participants in our rural study area described a social environment where syringe sharing was common and perceived as a widespread and even normative practice. However, one subset of participants chose not to share syringes at all. These participants shared two key characteristics: first, they perceived HCV as a real and avoidable risk, and second, they all had access to either an SSP or a pharmacy that sold syringes. This finding suggests that HCV prevention efforts in rural communities should not only involve expanding access to sterile syringe sources but also include interventions that promote in PWID a perception that HCV is preventable. Of course, changing PWID perceptions of HCV risk is not trivial. Some participants in our study only became concerned about HCV transmission via syringe sharing after becoming infected with HCV themselves. Research from the HCV and HIV literature suggests that perceptions of HCV, and eventually risk behaviors, can be changed by increasing PWID knowledge of HCV and their self-efficacy for safer injecting [30,31,32].

HCV interventions should also consider the importance of social connections among PWID. Our results reiterate the findings from prior studies that PWID rely not only on their knowledge of HCV transmission when making decisions about risk behaviors but also on emotions of trust and intimacy that serve as their own symbolic markers of disease risk [6]. We found the more intimate the relationship between two PWID, the more willing they were to share syringes with each other. This HCV risk mitigation strategy has been observed among various PWID populations and has been previously described as “discriminative” or “exceptional” sharing [33]. This is akin to the practice of “serosorting” described in the HIV literature, a prevention strategy where individuals choose sex partners and injection partners with concordant HIV status [34,35]. In an urban study of PWID attending opioid treatment programs, researchers used phylogenetic analysis to confirm that all HCV transmission linkages in their sample occurred among spousal or common-law partners [36]. As a participant in our rural study suggested, the social environment might be particularly salient in rural communities where PWID networks are smaller and perhaps more intimately interconnected than those in urban communities. Future research is needed to explore whether rural injection networks are at higher risk of syringe sharing within intimate relationships compared to urban injection networks.

Finally, our study highlights the importance of HCV education in mitigating injection equipment sharing practices among current PWID. A few of our participants described other PWID who shared syringes indiscriminately and were indifferent to the risk of acquiring HCV. Although these were virtually all secondhand accounts, participants in previous studies have described this same fatalistic attitude toward syringe sharing and HCV [6,37]. We did have one firsthand account of a PWID who, after acquiring HCV, developed a self-described “Well, I already got it” attitude and began sharing syringes frequently. Given that acquiring HCV motivated other participants to stop sharing syringes, several factors likely contribute to the development of an indifferent attitude toward syringe sharing and HCV risk. One factor is likely a social environment characterized by limited knowledge and inadequate education about HCV. In our study, we had participants who harbored misconceptions about HCV transmission and treatment, participants who described receiving inadequate HCV education at their time of diagnosis, if any, and others who remained confused about their own HCV antibody status. Researchers conducting qualitative research among PWID in both rural and urban settings have observed similar findings [6,12,38,39] and have suggested that confused and uncertain knowledge of HCV could work against the perception of HCV as preventable or treatable. Interestingly, one participant believed that indifference among a fellow PWID towards HCV was the result of their perception of HCV as curable, similar to other qualitative studies among PWID that have observed that the discernment of HCV as curable contributes to PWID perceptions of HCV as a less important and serious infectious disease threat compared to HIV [40,41]. However, unlike our study, participants in these studies did not make a direct connection between the perception of HCV as curable and syringe-sharing behavior. Although secondhand accounts should be interpreted with caution, our findings suggest that education regarding HCV may decrease injection equipment-sharing practices among current PWID.

This study has limitations. It is important to acknowledge that the data for this study were collected prior to the COVID-19 pandemic. Since then, there have been important changes to the public health context of injection drug use in our study area. Notably, SSPs in the study area responded to the pandemic by temporarily transitioning to an all-mobile distribution model, delivering sterile syringes and other harm-reduction supplies directly to PWID in the community. Harm reduction agencies adapted similarly in other rural communities [42,43]. Additionally, during the pandemic, a new SSP opened in one of the study counties in New Hampshire. To our knowledge, there has not been any research into the impact of the COVID-19 pandemic on PWID in our rural region of New England. In rural Illinois, PWID reported that the COVID-19 pandemic led to increased drug use (including new transitions to injecting drugs), increased drug use alone, more overdoses, and increased feelings of depression, anxiety, and loneliness [42]. However, PWID were also deeply appreciative and proud of harm reduction agencies for maintaining their services and came to rely on them as trusted sources of information during the pandemic. Future research in our study region is needed to determine whether the increased capacity for the delivery of harm reduction services during the pandemic has led to lasting increases in syringe access for PWID. Future research in rural New England should also explore the impact of pandemic precautions and social isolation on syringe-sharing behaviors and strategies among PWID. Finally, it is possible that the pandemic may have changed PWID perceptions of HCV, infectious disease testing, and HCV treatment.

Another limitation of this study is that our results may not be transferable to other rural counties or states beyond our 11-county study area in northern New England. Additionally, these interviews were not conducted with our specific research question in mind; despite the rich data in the interviews, it is possible we did not reach saturation. Nevertheless, ours is one of few qualitative studies to examine the HCV risk environment in a sample of rural PWID. Another strength is that our sample included participants from three different New England states, which allowed us to obtain a glimpse of how PWID and the HCV risk environments may differ between these neighboring states with varying harm reduction and political landscapes. However, given our relatively modest number of interviews, differences between the three states should be interpreted with caution.

## 5. Conclusions

Participants with a history of injection drug use living in rural NH, VT, and MA described risk environments characterized by limited access to sterile syringe sources that appeared to vary between the three states. This chronic lack of access to SSPs and pharmacies that sold syringes drove PWID to use informal syringe sources and contributed to widespread syringe sharing. However, features of PWIDs’ social environments, including perceptions of HCV risk, HCV status, interpersonal trust and intimacy, and misconceptions about HCV, also influenced syringe-sharing behavior. Efforts to prevent and eliminate HCV among rural PWID need to expand syringe access and cultivate the perception that the risk of HCV is real, serious, and preventable. Interventions should also consider PWIDs’ social connections as potential influences on syringe access and syringe sharing decisions.

## Figures and Tables

**Figure 1 viruses-16-01364-f001:**
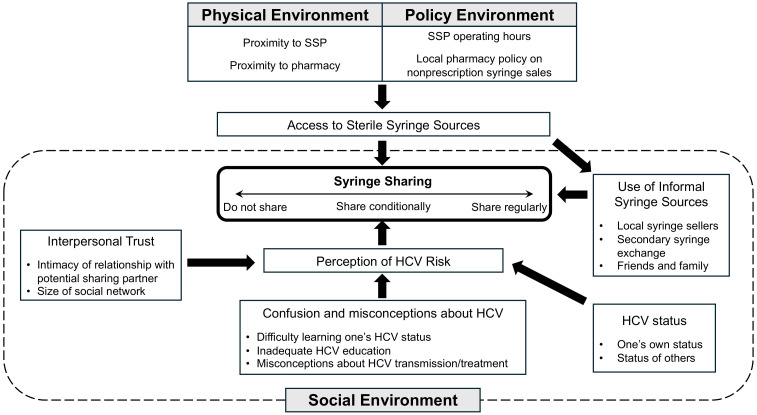
HCV risk environment among PWID in rural northern New England.

**Table 1 viruses-16-01364-t001:** Characteristics of semi-structured interview participants in rural New England, United States, 2018–2019 (n = 21, unless otherwise noted).

Characteristic	N (%)
**Residence**	
State	
Vermont	11 (52)
New Hampshire	6 (29)
Massachusetts	4 (19)
**Sociodemographics**	
Gender: women	11 (52)
Age (years): median (Q1–Q3) ^a^	29.5 (28–35)
Race/Ethnicity: non-Hispanic White ^b^	14 (93)
High school education or higher ^b^	12 (80)
Experienced homelessness (past 6 months) ^b^	8 (53)
**Criminal justice involvement**	
Incarcerated (past 6 months) ^b^	6 (40)
**Substance use**	
Injection drug use	
Currently injecting (past 30 days)	17 (81)
Not currently, but previously injected (past year)	4 (19)
Drug of choice ^b^	
Heroin	12 (80)
Fentanyl/carfentanil	3 (20)
**Infectious disease**	
HCV seropositive ^c^	12 (57)

^a^ n = 20; ^b^ n = 15. ^c^ Of the 12 HCV seropositive participants, 9 were determined to be seropositive using a rapid antibody test. The remaining 3 participants self-reported their positive HCV serostatus.

## Data Availability

The data generated in this study are available upon reasonable request from the corresponding author due to privacy reasons.

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
