# Peer review of "Syringe Access, Syringe Sharing, and Perceptions of HCV: A Qualitative Study Exploring the HCV Risk Environment in Rural Northern New England, United States"

_viruses, 2024, doi:10.3390/v16091364_

Round 1
Reviewer 1 Report
Comments and Suggestions for Authors
This is a qualitative study on an important issue of syringe access among rural PWID. The manuscript is well written and some interesting data are presented, but some methodological issue limit enthusiasm.
Line 51 - The authors state that there are few qualitative studies have explored the HCV risk environment among PWID in rural settings. However, there have been quite a few:
https://pubmed.ncbi.nlm.nih.gov/36641816/
https://pubmed.ncbi.nlm.nih.gov/35206421/
https://pubmed.ncbi.nlm.nih.gov/34996466/
https://pubmed.ncbi.nlm.nih.gov/34353216/
The introduction would benefit from more introductory information to set up their analysis.
Line 58 - The study should be reported in line with qualitative reporting guidelines such as COREQ.
Line 68 - There's no information on the methods surrounding sample selection - Was any purposive sampling conducted (e.g. for females, location)? Was it just a convenience sample? There's also not information on when recruitment and interviews were performed and over what time frame. These details are necessary to understand the policy landscape and timeline of HCV treatment availability.
Line 83 - Who was the survey instrument pre-tested with? This should be specified.
Line 113 - If the participants are from MA, NH and VT, were the interviews done remotely through teleconferencing? This isn't mentioned in the methods.
Line 162-165 - It's unclear why the authors deidentify this state when all others have been included.
Line 237 - How was it that all who were diagnose with HCV were able to obtain clean syringes. This seems unlikely with the material scarcity. There is potential for desirability bias here, which should be stated as a limitation.
Line 450 - There are data to suggest that such ties also occur in urban environments including phylogenetic confirmation of transmission along such lines. See for e.g. https://pubmed.ncbi.nlm.nih.gov/32150621/
Reviewer 2 Report
Comments and Suggestions for Authors
Thank you for the opportunity to review this manuscript reporting on a qualitative study exploring people who inject drugs' (PWID) experiences and perceptions of hepatitis C virus (HCV) and injection equipment sharing practices in rural northern New England in the United States. 21 interviews, conducted between April 2018 and August 2019, were analysed. The manuscript is well written and offers interesting and relevant findings. However, I have a number of comments.
Introduction
1. The introduction is very short and only includes 8 references. Some statements (e.g., Most of these new HCV infections are likely attributable to injection sharing practices, including sharing syringes and other drug paraphernalia) are not supported by citations.
2. The authors state "Nonetheless, few qualitative studies have explored the HCV risk environment among PWID in rural settings despite their disproportionate HCV burden" without providing any citations. I would appreciate an additional paragraph citing and discussing relevant qualitative studies that had been conducted to explore the HCV risk environment and/or perceptions of HCV in North America or high-income countries. I believe these studies should also be included in the discussion to situate and contrast the findings of this study within the context of global evidence.
Methods
3. The interviews were conducted pre-COVID 5 to 6 years ago (April 2018 to August 2019). There is no mention of the fact these interviews were conducted in a different public health context and how this context has evolved since then. I would appreciate this being discussed (and contextualizing these findings in the current public health landscape) and also brought up in the study limitations.
4. How was the interview guide developed? How many questions did it include? Was it based on a conceptual framework? How was it pre-tested?
Results
5. The information on the sample is very limited. Did all participants have stable housing? What types of drugs were injected? Any other information to provide some more context on participants?
6. "At one extreme were participants who did not share syringes at all." - How many participants fell in each category (sharing, not sharing, sharing only sometimes)?
7. The results are generally well presented. I would appreciate even more details and quotes for the different themes to provide some more substance to the paper (as now it feels a tad light).
Discussion
8. The discussion is very well written, and a strength of the paper. It would be improved by addressing the points discussed before.
Round 2
Reviewer 1 Report
Comments and Suggestions for Authors
Thank you to the authors for their revisions. I think the manuscript is improved considerably and I do not have any further comments on the updated version.
Reviewer 2 Report
Comments and Suggestions for Authors
Thank you to the authors for the revisions made to the manuscript. I appreciate the thoroughness with which my comments were addressed. All of my concerns have been resolved.